# Relevance of Human Papillomaviruses in Head and Neck Cancer—What Remains in 2021 from a Clinician’s Point of View?

**DOI:** 10.3390/v13061173

**Published:** 2021-06-18

**Authors:** Markus Hoffmann, Elgar Susanne Quabius

**Affiliations:** 1Department of Otorhinolaryngology, Head and Neck Surgery, Medical Faculty, Christian-Albrechts-University Kiel, D24105 Kiel, Germany; ElgarSusanne.Quabius@uksh.de; 2Quincke-Forschungszentrum (QFZ), Medical Faculty, Christian-Albrechts-University Kiel, D24105 Kiel, Germany

**Keywords:** HPV, outcome, smoking, co-morbidity, detection method

## Abstract

Human papillomaviruses (HPV) cause a subset of head and neck cancers (HNSCC). HPV16 predominantly signs responsible for approximately 10% of all HNSCC and over 50% of tonsillar (T)SCCs. Prevalence rates depend on several factors, such as the geographical region where patients live, possibly due to different social and sexual habits. Smoking plays an important role, with non-smoking patients being mostly HPV-positive and smokers being mostly HPV-negative. This is of unparalleled clinical relevance, as the outcome of (non-smoking) HPV-positive patients is significantly better, albeit with standard and not with de-escalated therapies. The results of the first prospective de-escalation studies have dampened hopes that similar superior survival can be achieved with de-escalated therapy. In this context, it is important to note that the inclusion of p16^INK4A^ (a surrogate marker for HPV-positivity) in the 8th TMN-classification has only prognostic, not therapeutic, intent. To avoid misclassification, highest precision in determining HPV-status is of utmost importance. Whenever possible, PCR-based methods, still referred to as the "gold standard”, should be used. New diagnostic antibodies represent some hope, e.g., to detect primaries and recurrences early. Prophylactic HPV vaccination should lead to a decline in HPV-driven HNSCC as well. This review discusses the above aspects in detail.

## 1. Introduction

A considerable amount of benign and malignant diseases are caused by infections with human papillomaviruses (HPV) [1]. The role of these viruses in the initiation of cutaneous and mucosal lesions of humans has first been described by Harald zur Hausen and his colleagues at the German Cancer Research Center, Heidelberg [2]. Specifically, and of pivotal importance, was the finding that HPVs are causally linked to the development of cancers of the cervix uteri [3]. Only after the HPV-associated carcinogenesis of cervical and other anogenital malignancies was established, has a link also been detected between benign and malignant mucosal neoplasms in the head and neck region: HPV6 and 11 sign responsible for the occurrence of recurrent respiratory papillomatosis [2] whereas predominantly HPV16 causes a subset of squamous cell carcinomas of the head and neck (HNSCC) [4,5,6]. Since the first hallmark publications on HPV-associated HNSCC in the late 1990ies and early 2000s [7,8,9,10,11], there has been a rapid increase in knowledge regarding HPV and, consecutively, its clinical relevance in HNSCC. Compared to cervical cancers where nearly 100% of patients are HPV-positive, with roughly 30–50%, only a subgroup of HNSCC [12] is caused by HPV [5,13], a fact that allows to compare HPV-driven and not-HPV-driven carcinomas and their characteristics such as biological behavior rather easily. Not HPV-driven HNSCCs can most likely be attributed to the consumption of tobacco and alcohol, the ingrediencies of which form classic risk factors for head and neck carcinogenesis [14]. Most intriguing, patients with HPV-driven HNSCC show significantly better progression-free and overall survival compared to patients with HPV-negative HNSCC [13,15,16]. As a result, knowledge of the HPV status in HNSCC has led to unparalleled clinical relevance, e.g., HPV status is an important prognostic factor, and is considered to possibly influence treatment decisions, in terms of therapy de-intensification [17]. However, this hope has recently been dampened by the negative results from two large clinical trials aiming to successfully treat patients with HPV-driven HNSCC with attenuated, i.e., de-intensified, treatment regimens [18]. Nevertheless, due to the described significant clinical relevance of the HPV status for HNSCC patients, the scientific community is intensely striving (i) to identify the natural history of HPV in the oral cavity followed consecutively by transition to active HPV infection which might lead to malignant transformation of infected cells, (ii) to establish the most reliable, yet easy to perform and inexpensive detection methods to (iii) determine true HPV prevalence rates of active—indeed driving—carcinogenic infections in different populations in the world, and finally (iv) to identify the subgroup of patients that might benefit the most from attenuated treatment regimens and HPV vaccination, respectively. This article is meant to provide an up-to-date overview of the aforementioned topics without claiming to be all encompassing. Rather, it is intended to provide the most accurate overall impression possible for head and neck oncologists in particular, but also for the entire HPV community, and to give a perspective what the most likely future status of HPV in head and neck oncology might be. 

The authors strive for the greatest possible objectivity here, but base their opinion on their own results, which have been determined in various studies on nearly 2000 tissue samples from approximately 1800 patients. In these studies, besides over 1100 HNSCC primary tumors (of which 350 were tonsillar carcinomas) [13,16,19,20,21], carcinoma of unknown primary syndromes [16,22], clinically inconspicuous mucosa partly from HNSCC patients, and non-neoplastic tonsils were analyzed [23,24]. The amount of data from a single university hospital in Germany generated in the same laboratory under comparable laboratory conditions and experimental methods (considering methodological changes that occurred over the period of almost 30 years) is certainly remarkable. Our expertise that has increased on the basis of our own studies in 30 years, however, allows a partly critical view of the presently supposed mainstream knowledge on HPV in head and neck oncology, such as our early criticism of the introduction of p16^INK4A^ immunohistochemistry as the sole detection method for HPV infections in oropharyngeal carcinoma in the current TNM classification (AJCC/UICC) [25,26]. Throughout, the authors aimed to provide scientific evidence to corroborate their partly critical or even contradictory view on of the mainstream knowledge regarding HPV-driven diseases of the head and neck.

## 2. Natural History

HPV-driven carcinogenesis starts with mucosal infection [27]. Regarding cervix uteri, it is well established that successful mucosal HPV infections lead via precursor lesions (former CIN I to III) to carcinomas [12]. There is a large amount of data on cervical cancers regarding HPV infection, latency, persistence, as well as clearance of the infection [1,12,27,28]. Anogenital HPV infections are sexually transmitted; consequently, correlated HPV-driven diseases are classified as sexually transmitted diseases (STDs) [1,12,27,28]. Comparable detailed data on HPV-associated SCCs of the head and neck are missing so far [29]. However, comparable knowledge has been gathered in natural history studies analyzing oral HPV prevalence, persistence, and clearance in the general population and specific subgroups such as HIV-positive individuals, here again specifically, men having sex with men [30,31,32,33]. In summary, such oral HPV natural history studies clearly show that persons with HPV infection of the oral cavity show a riskier lifestyle in terms of being (marijuana) smokers and/or show riskier sexual behavior (i.e., more sexual partners, performing oro-genital sex, etc.) [30,31,32,34]. Chaturvedi and co-workers [35] indicated in their NHANES-based natural history study of oncogenic HPV infections in the oral cavity that the distribution of oral oncogenic HPV infections in the US population parallels the incidence of HPV-positive oropharyngeal squamous cell carcinoma (OPSCC) at the population level. Infection and cancer were most common in men between 40 and 59 years who had never smoked or were former smokers. Just recently, a case control study on HPV-positive and -negative OPSCC was published, strongly proposing a significant correlation between riskier sexual behavior and HPV-positive tumors as well as between tobacco and alcohol consumption and HPV-negative tumors [36]. Thus, according to a considerable amount of data investigating this issue, it seems justified to suggest that HPV-associated diseases in the head and neck should be addressed as STD, also, with the latter certainly being correlated to riskier sexual behavior. Drake and co-workers [37] published only recently an HPV antibody study on 163 HPV-driven oropharyngeal cancer cases (OPSCC, here with HPV status also determined by p16^INK4A^ immunohistochemistry (IHC) and FISH) comparing these to 345 healthy controls concluding that the number of oral sex partners, timing, and intensity of oral sex are independent risk factors for HPV-driven OPSCC. The shown data clearly describe a difference in sexual behavior between the HPV-positive cancer cases and the control group. However, in a study conducted by us analyzing 106 questionnaires of patients with tonsillar SCC (n = 81) or with tonsillar hyperplasia [tonsillectomized due to sleep apnea syndrome (n = 25)], there could be no correlation detected between HPV status (determined by HPV DNA and RNA detection plus p16^INK4A^ IHC) of the investigated tonsillar tissues and sexual behavior of patients [29]. Intriguingly, among the individuals of the control group studied by Drake and co-workers [37] and tested for serum antibody levels, there indeed were no differences in sexual behavior between those with and those without elevated HPV antibody titers, thus being in accordance with our results [29]. In the study by Drake and co-workers, the study design did not include the comparison of HPV-positive and HPV-negative cancer cases or HPV-negative cancer cases to controls. 

Due to the limited sample size of our own study and the already mentioned limited evidence on the correlation between sexual behavior and HPV status in HNSCC patients, there is a clear need to increase the number of studies investigating the association between HPV-positive and HPV-negative cancer cases and sexual behavior, as well as other suggested risk factors such as other modifiable and non-modifiable risk factors, such as age, education, income, and so forth in different populations of the world and specifically outside the US. This notion is supported by substance abuse analysis as performed by Drake and co-workers [37] reporting that in both groups (HPV-positive cases and controls), over 90% of persons enrolled identified themselves as non-smokers. This is significantly different from our patients, where 70% and 50% of patients with HPV-undriven and HPV-driven cancers, respectively, do smoke, thereby representing a rather typical European study population [16]. Therefore, and due to other differences in study populations based on the geographical area the study population is derived from, data from US-American study populations might possibly not easily be transferred to other study populations. In fact, Shewale and co-workers [36] pointed out that due to lesser racial diversity, their hospital-based study conducted in Ohio is not even representative for the general US population. 

Additionally, there still is a need for natural history/cancer case studies investigating whether or not HPV-positive cancers arise from persons with incident/prevalent oral HPV infection detected in natural history studies, perhaps years earlier, and whether elevated antibody titers measured at any point in life are an indicator for (HNSCC-) carcinogenesis at a later point in life. Yet, researchers should separate results from natural history studies and studies on cancer patients. This adjustment could have significant impact on studies reporting that HPV acquisition and HPV-driven OPSCC are primarily due to sexual behavior. Moreover, our own results [29] and those from others [38,39] challenge the assumption that oral HPV infection exclusively is transmitted sexually. It repeatedly has been suggested that other, non-sexual forms of transmission possibly lead to oral HPV infection, specifically in terms of horizontal and vertical HPV transmissions at the time around of birth [38,39]. A more detailed critical view of the link between HPV natural history and HPV-associated carcinogenesis is given elsewhere [29]. There, we emphasize that in patient communication, it is important to stress that HPV is widespread and HPV infections can occur in sexually active people without causing a disease. Patient information should focus on the better prognosis of HPV-positive cancers and not on the potential association with sexual activity. 

## 3. Detection Methods

### 3.1. p16^INK4A^ Immunohistochemistry and the 8th Edition of the TNM Classification 

Several methodological approaches are currently available to determine HPV status in tissue specimens. Since the inclusion of p16^INK4A^ immunochemistry (IHC) in the 8th edition of the TNM classification (AJCC/UICC) for the determination of HPV status in oropharyngeal carcinoma, p16^INK4A^ analysis has become part of clinical routine. Since then, the WHO decision to recommend p16^INK4A^ IHC as the sole marker for the detection of HPV infection has been widely criticized and the challenges have been highlighted [25,40]. Briefly, a major point of criticism is that p16^INK4A^ is a cellular protein, which in a negative feedback mechanism, is enhanced in consequence of interference of HPV E6/E7 oncogene activity with cellular proteins on the cell cycle level (for mechanistic details see Refs. [41,42] and for active and inactive HPV infections, Section 4.4). Thus, p16^INK4A^ overexpression as a sign for viral activity is an indirect marker and by now it repeatedly has been shown that not by all other means has determined-active HPV infection, thus, biologically relevant, indeed show p16^INK4A^ IHC overexpression. To this end, we demonstrated in a first study comparing polymerase chain reaction (PCR)-based results of HPV DNA (HPVD) and mRNA (HPVR) presence with p16^INK4A^ expression levels that, up to 21.4% of 14 HPVD+/HPVR+ cases, did not show overexpression of p16^INK4A^ [43]. Moreover, we also detected tumor specimens with p16^INK4A^ overexpression that did not contain HPV DNA, thereby raising the first concerns regarding the validity of p16^INK4A^ as a sole marker for HPV positivity. Since then, several studies by us and others [13,16,23,44] demonstrate a clear discrepancy between p16^INK4A^ expression and HPV DNA/RNA, not only retrospectively, but also recently by us in a prospective study. 

However, despite the scarcity of evidence regarding its validity, p16^INK4A^ was deliberately included in the 8th edition of the WHO TMN classification [26]. The reason to include p16^INK4A^ into the current classification, which is after all valid worldwide, is that the method must be available and applicable also worldwide. Moreover, due to the fact that only oropharyngeal SCC are addressed and since it is established that the proportion of active HPV infections among HPV-positive SCC of the tonsils is almost 100% [21], at the time of designing the TNM classification, the authors assumed to be safe in their recommendation. Only later did data of the aforementioned discrepancy between true HPV status and p16^INK4A^ expression and, moreover, on the differences in HPV susceptibility between tonsillar SCC and oropharyngeal SCC emerge (see also Section 4.3) [13,40]. Therefore, it might be expected that the present 8th edition of the TNM classification will be up-dated rather soon. 

At this point, we just want to briefly mention the fact that we and others could show that p16^INK4A^-positive, but HPVD- and HPVR-negative, cases show better overall survival compared to negative cases for all three parameters [13,45]. Accordingly, p16^INK4A^ overexpression seems to be associated with a survival advantage, which, however, seems to be HPV independent.

In this context, the fact that p16^INK4A^ positivity is virtually synonymous to HPV positivity in routine clinical practice is problematic. This superficial approach carries the additional risk that patients may be misclassified based on the p16^INK4A^ results. Precision and care are required here, and it needs to be emphasized that the TNM classification has exclusively prognostic, but not therapeutic, intent. The latter has been interpreted slightly distortedly by the HNSCC/HPV community, since rather accidentally at the same time of publishing the new TNM classification, prospective clinical trials investigating treatment de-escalation in HPV positive patients were launched [17]. Currently, it cannot fully be ruled out that due to the latter, HPV or rather p16^INK4A^ status already has influenced treatment decisions with the chance that based on misclassifications as described above, in some cases, patients have been inappropriately treated with a de-escalated treatment regimen.

### 3.2. p53—Wild-Type or Mutant

Tumor suppressor protein p53, encoded by the TP53 gene, is upregulated and leads to cell senescence when DNA damages are detected, and coordinates (radiation-induced) DNA repair. It acts in combination with specific cyclin-dependent kinases to cause cell cycle arrest at the G1 phase to allow for DNA repair or apoptosis, and checks DNA integrity before initiating cell division. TP53 mutations are highly prevalent in OPSCC, unrelated to HPV infection but driven by mutagenic substances present in tobacco and alcohol [46]. Accordingly, it is well established that tobacco/alcohol-associated carcinogenesis is correlated to p53 mutations whereas in HPV-driven HNSCCs in the majority of cases, wild-type (wt) p53 can be detected [47]. Therefore, in smoking HNSCC patients, which seem to be more common in Germany than in the USA (see Section 4), for instance, it might be helpful to distinguish via p53 analysis between tumors that rather are HPV-driven (wt-p53) or indeed associated with tobacco-associated carcinogenesis (mutant p53) [48,49]. Hence, and to overcome the aforementioned problem correlated to p16^INK4A^ IHC for HPV detection as described in 3.1, Benzerdjeb and colleagues [47] conducted a study on 110 OPSCC combining p16^INK4A^ IHC and p53 analysis in terms of detection of wt-p53 versus mutant p53. For p16^INK4A^-positive or p16 ^INK4A^-negative/wt-p53 cases (n = 63), DNA in situ hybridisation for high-risk HPV was performed, and if negative, the HPV status was controlled by HPV DNA PCR (n = 19). It could be concluded that p16^INK4A^ positivity combined with the presence of wt-p53 is a strong predictor for HPV positivity in OPSCC, while p16^INK4A^ protein immunopositivity in conjunction with a mutant-type p53 staining rather is associated to HPV-unrelated tumors. In routine clinical practice and when PCR-based HPV detection methods are not applicable, it therefore seems advisable to perform p53 IHC analysis in addition to p16^INK4A^ IHC to significantly reduce the rate of misclassification when p16^INK4A^ IHC is applied alone. Whether and to what extent the latter also holds true for non-OPSCC HNSCC should be addressed in further analysis on the matter. Nonetheless, the combination of IHC for p16^INK4A^ and p53 will add to more precise detection of HPV-driven HNSCC, most likely when additionally combined to PCR-based HPV DNA and mRNA detection methods, the latter forming the gold standard in HPV detection.

### 3.3. Gold Standard in Detection of Active HPV Infections

The gold standard for the detection of biologically active, thus carcinogenic, HPV infections is the analysis of E6/E7 mRNA by RT PCR, with very high sensitivity and specificity [36,43]. Compared to p16^INK4A^ IHC, this method might have the disadvantage that it requires sample preparations of the tissue DNA and mRNA. However, kits for nucleic acid extraction, also for FFPE samples, are commercially available and are easy to use. The detection of HPV oncogene E6/E7 transcripts is of utmost importance since HPV-driven carcinogenesis critically depends on the interaction of the HPV E6 and E7 oncogenes with cell cycle proteins (for mechanistic details, see Refs. [41,42]). Based on the analysis of over 1100 HNSCC specimens, we strongly advocate, wherever possible, to analyze PCR-based HPV DNA followed by mRNA expression analysis and correlate mRNA results to p16^INK4A^ IHC to achieve the highest accuracy. We repeatedly have carried out studies following the latter recommendation and believe to have generated the most reliable and reproducible results with the highest avoidance of misclassification. It seems that a growing number of scientists rather shift from primarily p16^INK4A^ IHC-based to mRNA-PCR-based detection methods, as only recently shown by Shewale and co-workers still addressing this approach as gold-standard [36].

Although detection of E6/E7 transcripts was an inclusion criterion for a meta-analysis performed by Prigge and co-workers [45], the authors only assessed the accuracy of HPV DNA detection by means of PCR or (fluorescence) in situ hybridization [(F)ISH] alone or in combination with p16^INK4A^ IHC, concluding that high sensitivity but only moderate specificity is achieved when p16^INK4A^ IHC and HPV DNA PCR are solely used to detect active HPV infections. However, when combining these two methods, specificity can be significantly optimized without affecting sensitivity. An international multi-center study on the worldwide incidence of HPV-associated HNSSC [50] showed that with increasing precision in HPV detection in terms of single or combined detection methods, the HPV attributable fraction of HNSCC decreased significantly. Vice versa, low precision in HPV detection increases the proportion of false HPV positives, allocating patients into de-escalating treatment arms, possibly resulting in insufficient treatment and worse survival.

### 3.4. (Fluorescence) In Situ Hybridization for HPV Detection—HPV (F)ISH

The development of the in situ hybridization (ISH) technique by Gall and Pardue [51] enabled the detection of specific DNA and RNA sequences, first using radioactive probes and later on, the method was adapted by using fluorescent probes for hybridization (FISH). Probe hybridization to DNA or RNA sequences requires the presence of single-stranded DNA or RNA, and for DNA detection, a denaturation step is needed, resulting in the destruction of the RNA. RNA (F)ISH does not provide conditions for the detection of DNA. Since RNA is already single-stranded, there is no need for denaturation. Combining RNA and DNA (F)ISH within one experiment is therefore hard to achieve and requires highly specialized protocols [52] and references therein]. These methods are, to the best of our knowledge, up to now, not tested in HNSCC samples.

For HNSCC samples, kits for separate identification of HPV DNA and HPV RNA by means of ISH or FISH are commercially available, but are still rather laborious. FISH analysis needs a special (fluorescence) microscope to visualize the HPV-infected cells. ISH, on the other hand, uses chromophores that can be visualized with a normal microscope. Both methods can determine the presence of HPV-infected cells directly on tissue slides by means of microscopy. Moreover, the methods are able to differentiate between tumor and surrounding non-tumoral tissue. In this context, it is important to mention that Chi and colleagues [53] demonstrated that overexpression of HPV DNA and HPV RNA only occurs in the tumor tissue. Therefore, this refutes the criticism that PCR-based HPV DNA and RNA detection methods are mostly performed on tissue blocks containing tumor and surrounding non-tumoral tissue, with the latter being occasionally criticized to falsify results when examining both tissue types at the same time.

RNA ISH has higher sensitivity compared to DNA ISH, likely reflecting the fact that although viral DNA can be present in low copy numbers, transcription is a natural amplification step that results in high levels of viral mRNA. A study comparing DNA ISH and RNA ISH on head and neck surgical specimens, using detection of HPV by PCR as the gold standard, showed that RNA ISH was 91% sensitive compared to 65% for DNA ISH [54]. A separate study of another commercially available RNA ISH detection system showed a sensitivity of 97% and specificity of 93% when compared to quantitative reverse transcription PCR as the gold standard [55]. 

Since none of the above-mentioned HPV detection methods reach the same specificity and sensitivity as the PCR (DNA and RNA)-based methods, these methods should still be considered as “gold standard” to determine HPV infections in the context of epidemiological and natural history studies. (F)ISH appears to be an excellent method when solving basic research-related questions relating, for example, to the precise location of the infected cells.

### 3.5. Novel Blood Based HPV Tumor Marker—Recent Innovation in HPV Diagnostics

The value of measuring antibodies against HPV in sera of HNSCC patients has already been introduced some 20 years ago [56,57]. However, up to date, this method has not found its way into (routine) diagnostics, but is included into, for example, natural history studies. The latter perhaps is due to the fact that data on sensitivity and specificity are not as solid yet, and its applicability as a tumor marker remains questionable [58]. In a study investigating 161 OPSCC patients with known HPV status, Lang Kuhs and co-workers [59] performed HPV16 E6 antibody analysis of pre-treatment and partly post-treatment sera and could not detect a significant decrease of antibody levels in post-treatment sera and levels were not associated with the risk of recurrence. However, pre-treatment HPV16 E6 seropositivity was associated with an 86% reduced risk of local/regional recurrence. The authors concluded that HPV16 E6 antibodies may have potential clinical utility for the diagnosis and/or prognosis of HPV-driven OPSCC. However, Weiland and co-workers [60] published only recently a study utilizing a novel HPV16-L1 DRH1 epitope-specific antibody applied in a multi-center study analyzing sera from 1486 patients suffering from various HPV-associated diseases. Here, the authors showed a decline in post-treatment antibody levels and concluded that HPV16-L1 DRH1 epitope-specific antibodies are linked to HPV16-induced cancer. As a post-treatment biomarker, the assay allows independent post-therapy monitoring as well as early diagnosis of tumor recurrence. An area under the curve of 0.96, as determined by Weiland and coworkers, indicates high sensitivity and specificity for early detection of HPV16-driven disease. Hence, further studies evaluating this antibody seem rather promising.

## 4. Epidemiology

### 4.1. Geography and Smoking

The aspects of global HPV epidemiology are multifaceted and challenging. One of the reasons for the latter is the diversity of methodical approaches to detect HPV-positive, specifically HPV-driven cases as described in Section 3. Another aspect is the still not fully understood diversity in HPV prevalence rates in head and neck cancers in various populations tested throughout the world [13,61]. The patients’ smoking habit possibly is one major factor responsible for the geographic impact on HPV prevalence rates since it is well established that patients with HPV-positive and HPV-negative cancers predominantly are non-smokers and smokers, respectively [13,15,16]. Therefore, in countries with a comparably small proportion of smokers in the (study) population as is, for instance, regularly described for US-American cohorts among others by Drake and co-workers [37], with even less than 10% active smokers (see Section 2), the burden of HPV-driven OPSCC might be higher due to the smoking habit of the (study) populations. Indeed, the OPSCC HPV prevalence rate in the USA and Sweden is estimated to be approx. 60 and 70%, respectively [61]. In the USA and Sweden, the proportions of smokers in the general population, with reported 14% (Centers for Disease Control and Prevention, 2019) and 7%, respectively, are low (statista.com, access data: 30 May 2021). Sweden is the country with the lowest rate of smokers in Europe. In Germany, with 24% of smokers in the general population (Eurostat) the HPV prevalence rate is reported to be approx. 40% in OPSCC. A recent meta-analysis on the global prevalence of HPV-driven OPSCC, following the ASCO guidelines [61], describe a pooled global prevalence of HPV-driven OPSCC of 45% with the highest rates in New Zealand (75%, proportion of smokers 2018: 13%, trend declining [www.stats.govt.nz, access data: 30 May 2021]) and the lowest rates in Brazil (11%). However, the alignment of various global HPV prevalence rates to official public governmental statistics on the burden of smokers in the population as performed above should not be conducted carelessly, since the proportion of smokers in the entire population might significantly differ from the proportion of smokers in the studies themselves. As mentioned, the official proportion of smokers in Germany and the USA has recently been estimated to be 24 and 14%, respectively, whereas in one of our own studies, the proportion of smokers was 50 and 70% in HPV-positive and -negative cases [16], respectively, and Drake and co-workers [37] reported on 7% of smokers in their study. This discrepancy certainly is due to the preselection of cases in HNSCC studies, the latter per se being more intensively associated to substance abuse when compared to the general population [62]. Meanwhile, the burden of smokers is declining worldwide. In Brazil, for instance, between 1990 and 2017, smoking in the population decreased from 35.3 to 11.3% [63] and still is declining [64]. Therefore, determination of the proportion of smokers (and other confounding factors associated with HNSCC carcinogenesis) should be integrated in every study design investigating HPV epidemiology. 

In this review, it should be mentioned only briefly that, so far, the background for the described interaction between smoking habit and HPV status is not fully understood. However, based on our own results on more than 1000 patients (which add to the by-us investigated cases summed up in the introduction) and supported by two US-American studies, we postulated the following hypothesis, perhaps elucidating this interaction: smoking leads to increased SLPI and AnxA2 expression in mucosal tissues with significant SLPI excess. SLPI binds to AnxA2, which consecutively inhibits the binding of HPV, if present, to AnxA2. HPV binding to AnxA2 is crucial for successful HPV infection of mucosal cells. Conversely, in non-smokers with significantly higher levels of AnxA2 compared to SLPI, HPV can bind more readily to unoccupied—non-SLPI-bound—AnxA2, and successful infection of cells is likely. Our various studies on this topic are summarized in [65] and data from a prospective study performed only recently are depicted in [23].

### 4.2. Alcohol Consumption 

Smoking habit most likely influences HPV prevalence rates in combination with other factors of interest, such as, among others, alcohol uptake and sexual behavior, perhaps even in a synergistic way [66]. The impact of sexual behavior already has been discussed in Section 2. Studies on HPV prevalence rates and alcohol consumption are sparse and usually combined with smoking and socio-economic factors such as income, education, and race [67]. Assessment of alcohol consumption and herewith the related study results often are less reliable since there is not an as-clear measure as, for example, pack years for smoking. Due to the fact that alcohol abuse often is associated with smoking, the impact of alcohol itself on HPV infection (rates) seems hard to elucidate. In an earlier study on smoking, co-morbidity, and treatment compliance [68], we discussed that smoking and drinking rely on the patients’ subjective impressions, which specifically for drinking is not easy to specify in terms of defined measures. For smoking, the measure “pack years” is easy to handle for patients and clinicians. However, such well-defined measurements for alcohol consumption are not available. Aarstad and co-workers [69], for instance, defined a person as a “regularly drinking person” when the latter reported to have more than two drinks a week. 

### 4.3. Anatomical Tumor Site 

Another major aspect influencing the topic of epidemiology is the diversity of primary tumor sites in the head and neck region. So far in this article, we predominantly addressed head and neck squamous cell carcinomas (HNSCC) as a whole and partly addressed oropharyngeal (OP)SCC. Except for SCC of the nasopharynx, the various anatomical tumor sites all show a subgroup of HPV-positive and even HPV-driven cases according to the criteria addressed in Section 3. In a multi-center study, with all the analysis being performed in a single laboratory (cases, n = 307), we showed the rate of HPV DNA positive cases in the larynx, hypopharynx, oral cavity, and oropharynx other than tonsil to be 9.1, 5.2, 5.1, and 15%, respectively, and the rate of HPV-positive tonsillar SCCs (TSCC) to be 43.7% [13], with HPV16 being the genotype detected in approx. 95% of all cases, followed by HPV18, and 33. These HPV DNA prevalence rates are in agreement with results of earlier own studies addressing the issue of anatomical tumor sites and HPV infection [19,43,70,71]. Pooling the cases of those studies (cases investigated, pooled n = 350), the determined HPV prevalence rates in SCC from the Waldeyers’ tonsillar ring (base of tongue (lingual) and palatine tonsil), the oropharynx other than tonsils, oral cavity, larynx, and hypopharynx were 56, 13, 28, 24, and 28%, respectively [13]. Thus, the by-us continuously applied categorization into the various tumor sites, specifically the separation between oropharynx other than tonsil and TSCC, is extremely precise and, moreover, gets to the core of the matter: the tonsils have the highest susceptibility towards HPV infections [10,72], as clearly depicted by the site-specific HPV prevalence rates shown above. On the other hand, OPSCC other than tonsillar SCCs show HPV prevalence rates as low as or even lower than the other anatomical regions of the head and neck [13]. In the majority of studies cited here except for ours, such a categorization has not been performed, but all sites of the oropharynx, tonsillar and non-tonsillar, have been examined together. Thus, corresponding to the number of cases counted as oropharynx, yet being derived from oropharyngeal sites other than the tonsils, the detected HPV prevalence rate will be either higher or lower than should be expected when exclusively tonsillar SCC would have been analyzed. In summary, it is important to keep in mind that there are two types of tissues in the head and neck region: (1) the lymphoepithelial tissue of the tonsils (both lingual and palatine tonsils) specifically affected by HPV-driven carcinogenesis due to the already mentioned higher susceptibility towards an HPV infection, and (2) the “normal” respiratory mucosa of the upper aerodigestive tract, which comprises the majority of anatomical sites [oral cavity, larynx, hypopharynx, and oropharynx other than tonsils (posterior wall of the oropharynx, soft palate, and posterior palatal wall)]. Since nomenclature regarding oral (OSCC) and oropharyngeal SCC (OSCC and/or OPSCC) is sometimes used in an inconsistent and confusing manner, we recommend to indeed separate tonsillar SCC (TSCC) from the other sites and additionally precisely indicate how the other non-tonsillar tumor sites are defined and which abbreviation is used. 

Probably due to the significantly higher HPV prevalence rates described for TSCC and OPSCC with a supposedly higher clinical relevance, a clear trend has developed worldwide to examine almost exclusively oropharyngeal SCC instead of analyzing all tumor sites from the head and neck region for HPV infections. Considering the worldwide incidences of SCC in the various anatomical head and neck tumor sites, numbers of HPV-driven tumors of all non-tonsillar tumors reach approximately 50% of HPV-driven TSCC. The latter is based on the overall significantly higher incidence of non-tonsillar SCC in comparison to tonsillar SCC. Cramer and co-workers [73] estimate the worldwide annual incidence of oropharyngeal SCC to be 80,608 cases, compared to 358,846, 177,422, and 92,887 for SCCs of the oral cavity and lip, larynx, and hypopharynx, respectively; resulting in 629,155 non-tonsillar SCCs annually. Based on the results of our multi-center study [13] reporting on 7.6% HPV-DNA-positive cases among the non-tonsillar cases, of which 38.5% carry active HPV infections (see Section 4.4), it can be estimated that 18,409 non-tonsillar SCC are HPV-driven. With 43.7% HPV DNA positives among the tonsillar SCC and 96.6% showing viral activity, HPV-driven tumors sum up to 34,028 cases annually worldwide. Thus, the total annual number of HPV-driven head and neck cancers sums up to 52,437, of which about one-third are non-tonsillar. Results of recent studies show that HPV-driven hypopharyngeal and laryngeal carcinomas also have better survival rates compared to HPV-negative ones. Thus, the biological behavior of HPV-driven non-tonsillar carcinomas also appears to be different from HPV-negative ones. The latter should not only be considered in scientific and clinical studies, but also in cost-benefit calculations, for example, when evaluating vaccination strategies.

### 4.4. Virus Activity

As indicated in Section 3, a precise HPV detection method should be able to reliably distinguish between HPV infection either with or without activity, since only active HPV infections are carcinogenic and are responsible for the biological behavior of tumors. As also outlined in Section 3, we believe that the use of a combination of different detection methods, such as HPV DNA and RNA PCR combined with p16^INK4A^ and perhaps p53 IHC, may be best suited to fulfill this requirement [47]. Indeed, it is of crucial importance to identify the biologically active, i.e., causative HPV infections since sole detection of HPV DNA “presence” does not justify the claim that the tested (malignant) lesion is HPV-driven. Therefore, only patients with active HPV infection, thus, uttermost likely HPV-driven cancers, will show the biological characteristics such as better survival despite higher disease burden, most likely of the lateral neck. We and others have shown that the proportion of active HPV infections, among those being HPV-DNA-positive in the first place, is approx. 96.6% and 7.6% in tonsillar and non-tonsillar cases, respectively; however, for the latter with only limited data [21,43,50,74,75]. When looking at different studies on HPV, it is therefore important to ask what the intention of the study was and whether or not any viral infection activity was shown at all. For natural history studies, and for studies investigating HPV transmission, it is negligible to determine whether or not HPV DNA presence, i.e., the infection, indeed represents an active HPV infection. However, to determine HPV-attributable cancer burden, viral activity must be detected, i.e., the underlying carcinogenicity of HPV infection. The latter specifically holds true when in future studies treatment regimens should be attenuated according to the HPV status. 

## 5. Treatment and Outcome

Treatment options for HNSCCs are surgery alone or in combination with risk-adapted adjuvant radio(chemo)therapy (R(C)T) and primary R(C)T [76,77]. Independent of the applied treatment, HPV-driven HNSCC, especially TSCC, show significantly better survival rates compared to HPV-negative HN/TSCC [13,15,16]. This holds true even though HPV-driven cancers present with higher burden of disease, specifically of the lateral neck (higher N category) accompanied by rather small primary tumors (small T category) [76,77]. Due to this most intriguing instance with highest clinical relevance, HPV status in OPSCC has been included in the 8th edition of the TNM classification (AJCC/UICC) [26] to reflect the favorable prognosis of HPV-positive TSCC. On the other hand, numerous clinical studies have been initiated to test whether the same outcome can be achieved with de-escalation of therapy in HPV-driven carcinomas [17]. The first two non-inferiority trials that substituted cisplatin for cetuximab in the trial arm for R(C)T of HPV-positive patients for the purpose of de-escalation were negative [18]. According to one of our previous studies, it might well be that these de-escalation studies did fail because of the enrollment criteria. Based on this study, only non-smoking HPV-positive patients, showing 100% survival after 10 years, might have qualified for such a de-escalation study [13]. However, only 10% of TSCC the patients analyzed by us belong to this group, raising the question if such de-escalation studies are really worth the while and not possibly inflicting more harm than good on some patients. 

In 2018, we published a study on the survival of 126 TSCC cases which initially were all surgically resected and treated risk-adapted with or without adjuvant R(C)T [16]. Overall (OS) and progression-free (PFS) survival of the analyzed patient population was excellent, regardless if stratified for the various parameters (i.e., HPV, smoking, age, gender), and could very well be explained by the performed surgical tumor resection prior to risk-adapted adjuvant treatment or surveillance. Therefore, it seems justified to upfront resect tumors surgically when respectability is given instead of routinely performing primary RCT for all tumor stages in case of HPV positivity or even sole p16^INK4A^ overexpression. This statement seems to be challenging since in some countries of the Western World, there appears to be a paradigm to treat this tumor entity by primary RCT; this however, lacks clinical evidence based on prospective clinical trials.

In agreement with studies by others [15], several own studies strongly corroborate the positive impact of HPV in head and neck cancer with significant superior OS and PFS for patients with HPV-positive tumors when compared to HPV-negatives. The latter is true (i) for all HNSCCs including laryngeal and hypopharyngeal SCC [78,79], yet with some conflicting data [80,81], but specifically for TSCC, and (ii) for HPV DNA positives but specifically for those cases with active HPV infections in terms of detection of viral mRNA. Prognosis of survival, however, is not solely correlated with the infection with HPV but additionally depends on smoking habit and the presence of co-morbidities with poorest survival rates for HPV-negative smokers and/or patients with co-morbidities [4,5,8,9,10,11].

Analyzing the association of smoking habit, co-morbidity, and achievement on planned R(C)T dosage in 643 HNSCC patients, we only recently confirmed the significant association between increased tobacco consumption and presence of co-morbidities (applying various co-morbidity indices as the CCI); however, we did not see a correlation between smoking and/or co-morbidities and an impaired achievement of R(C)T dosages [68,82]. Moreover, we found that former smokers showed survival rates as bad as active smokers. The latter, however, could only be attributed to those former smokers with co-morbidities since the former smokers without co-morbidities survived as well as never smokers without co-morbidities. Likewise, never and active smokers with co-morbidities showed inferior survival rates in comparison to never and active smokers without co-morbidities. Therefore, in survival estimates, the presence of co-morbidities seems to be more important than the smoking habit itself. In the 2018 study investigating the influence of smoking on HPV-positive and HPV-negative TSCC, we concluded that the positive impact of HPV infection on survival was fully jeopardized by a positive smoking history. Based on the aforementioned data, however, this might only be true for those patients with co-morbidities since the negative effects of smoking were in particular evident in patients with co-morbidities. Vice versa, in the absence of any co-morbidity, the effect of smoking appears to have only a minor impact on survival.

In conclusion, future studies on HPV and survival in HNSCC should, along with substance abuse, additionally integrate the presence of co-morbidities with the latter being of higher relevance than smoking, gender, and age. Results of de-escalation studies, for instance, might be negatively biased by imprecisely applied detection methods (i.e., sole p16^INK4A^ IHC, see Section 4) and missing or incongruent stratification for parameters impacting survival rates such as co-morbidity or smoking habit. Due to the growing amount of studies underlining (i) the necessity of precision when active HPV infections ought to be detected and (ii) the impact on survival that other factors such as co-morbidities and smoking might have, and most important (iii) the to-date unsuccessful attempt to treat patients with HPV-driven cancers in de-escalated treatment regimens, lead to the fact that US-American [83] and European [84] treatment guidelines do not differentiate between HPV-positive and HPV-negative cases. Therefore, treatment of patients with HPV-driven OPSCC is similar to that with HPV-negative OPSCC, except in the context of clinical trials.

## 6. Vaccination

Prophylactic vaccination against HPV is part of vaccination programs or recommendations throughout the world, initially only including females. However, to-date, in most countries, it has been expanded to males. The latter is due to benign and malignant diseases which are caused by HPV and to a substantial degree also affect the male population, such as, for instance, condyloma accuminata and HPV-driven anal SCC, which even is a marker cancer for HIV-positive males [85]. Last but not least, knowledge on HPV-driven HNSCC has contributed to the integration of males into the various vaccination strategies since it has been realized that HPV-driven OPSCC are rising in incidence [[75] and Refs. therein], and that with a male:female ratio of 4–5:1 [13,16], predominantly male patients suffer from these cancers. Hence, cost-effectiveness analysis in terms of HPV vaccination shifted toward including males into vaccination programs [86,87,88].

Since the onset of vaccination strategies, a significant decline of specifically benign and premalignant lesions of the female anogenital tract was recorded. This could be expected due to pre-onset studies on vaccination efficacy [89]. Hence, HPV vaccination has led to a dramatic decline in anogenital warts and other disease incidences in populations that have achieved high vaccination rates [86,90]. Similar, with the increasing utilization of the 9-valent and quadrivalent HPV vaccine in Australia, Benedict and Derkay [91] have seen a significant decrease in the incidence of recurrent respiratory papillomatosis. Preliminary data from the US show a similar trend of decreased incidence after implementation of vaccination. For HPV-driven cancers, the follow-up time within national HPV vaccination programs is still too short to evaluate the effect [86], although a positive trend has recently been demonstrated in one of the major HPV vaccine trials [92]. Hence, it is expected that HPV vaccination could eradicate cervical cancer within decades in those countries with high coverage such as Australia and the United Kingdom [93].

Therefore, HPV vaccination appears promising to reduce HPV-driven cancers of the head and neck in the future due to vaccination. However, effectiveness of HPV vaccination strategies strongly depends on herd immunity [94,95], which again depends on population uptake of vaccination, with the latter showing significant differences between countries [96,97,98]. These differences are based on various vaccination policies, population attitudes and concerns [97,98], and, moreover, on the negative impact that the COVID-19 pandemic has on the latter as only reported briefly [96]. However, due to its effectiveness also against HPV-driven lesions in the head and neck, we strongly advocate for HPV vaccination in a populational-based manner and suggest vaccination programs rather than recommendations to reach high vaccination coverage. With a high vaccination coverage rate (probably greater than 70% of the population), it can be achieved that not only vaccinated but also non-vaccinated persons benefit from the vaccination program due to the effects of herd immunity [99]. Thus, even non-vaccinated persons would present a decline in HPV-associated diseases. The goal of achieving high vaccination rates also with regard to HPV is accordingly worthwhile.

## 7. Conclusions and Future Directions

A significant subset of HNSCCs is caused by an infection with HPV16. It is important to distinguish between tonsillar and non-tonsillar SCCs (including oral cavity, hypopharynx, larynx, and oropharynx, except tonsils), as the HPV-attributable fraction (i.e., HPV-triggered carcinogenesis and tumors that are HPV-driven) of these can be estimated to reach approx. 50 and 7%, respectively. With an estimated global overall number of HPV-driven cancers of 600,000 cases annually, the subgroup of HPV-driven HNSCC with approx. n = 52,000 represents 9% of all HPV-driven cancers (HPV-driven TSCC annually, n = 34,400). Patients with HPV-driven HNSCC show significantly superior survival rates, which specifically holds true for HPV-driven TSCC, with these patients showing a 10-year overall survival rate of 100% when (i) without smoking history and co-morbidity and (ii) when treated in a non-de-escalated treatment regimen. Prevalence rates as well as outcome differ, however, significantly between study populations and countries due to not-well-understood geographical differences and differences in study designs (applied detection method, stratification for impacting factors such as smoking habit and/or co-morbidities). Depending on whether or not patients have a smoking habit and are with or without co-morbidities, the subgroup of patients that will benefit from de-escalated treatment regimens in terms of reduced therapy-related morbidity appears rather small after all. Although the goal of lowering morbidity is important, it seems questionable whether de-escalation studies indeed are appropriate in light of today’s more gentle treatment modalities, including, for instance, intensity-modulated radiotherapy and in light of the excellent survival of patients with HPV-driven SCC when treated un-de-escalated. In this context, it again needs to be stressed that the literally down-grading of p16^INK4A^-positive OPSCC in the current TNM classification (AJCC/UICC) exclusively has prognostic intent and the application of de-escalated therapy, without any exception, should only be applied in clinical trials. Nevertheless, the detection of HPV infections continues to be of great importance: it should be clarified how exactly infection per se and transmission occur and which mechanisms and influences, e.g., immunological ones, finally lead to carcinogenesis. An intense and consequent comparison of characteristics of patients with HPV-driven or HPV-negative cancers may contribute to the identification of such mechanisms. For both intentions, novel detection methods with respect to, for instance, liquid biopsies might be useful in future. Furthermore, describing the molecular biological differences between active and inactive HPV infections in SCC may reveal deeper insights into HPV-related carcinogenesis. Knowledge of such mechanisms may eventually help to better understand the carcinogenesis of HPV-unrelated cancers, i.e., most likely tobacco- and alcohol-associated head and neck cancers, thus opening the chance to gain insights that possibly could benefit the large HPV-negative majority among HNSCC patients. Additionally, more data is needed to corroborate the need for global sufficient vaccination policies to finally overcome HPV-related diseases.

## Data Availability

Not applicable.

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
