# Peer review of "Relevance of Human Papillomaviruses in Head and Neck Cancer—What Remains in 2021 from a Clinician’s Point of View?"

_viruses, 2021, doi:10.3390/v13061173_

Round 1

Reviewer 1 Report

In this review Authors described in detail all aspects of HPV-related HNSCC.

The knowledge about HPV and its clinical relevance in caners has been rapid increase.

Therefore the Authors  emphasize the importance of new diagnostic methods.  

p16INK4A immunochemistry, analysis of E6/E7 mRNA by RT PCR, HPV16 E6 antibodies may have potential clinical utility for the diagnosis and/or prognosis of HPV driven OPSCC.

The Authors rightly note that high vaccination rate (greater than 70% of the population) can be benefit  not only for vaccinated but also non-vaccinated persons. 

In conclusions, the authors highlight the directions for further research -how exactly infection per se and transmission occur and which mechanisms and influences, e.g immunological ones, finally lead to carcinogenesis.

More studies are needed to better understand the HPV-associated carcinogenesis. The manuscript is interesting, summarizes the existing knowledge and sheds light on new aspects of HPV-connected cancers. The literature is richly cited.

Author Response

Thank you for the kind comments.

Reviewer 2 Report

In the manuscript by Hoffman and Quabius, the authors review the current knowledge about HPV in head and neck cancer. The review is well written, informative and comprehensive.

There are only a few minor points that could be improved. The manuscript could also slightly benefit from some language/style changes and proofreading but the message is clear to read despite this.

P2 L50 Naming HPV status as the „most important prognostic factor“ might be too strong. One would assume that HPV doesnt outweight the prognostic significance of high overall TNM stage. Please adjust.

P3 L103-107. The supporting data of the referenced study (36) clearly specify the inclusion of oral and oropharyngeal cancer patients. While the abbreviation used therein („OSCC“) is suboptimal and confusing the supplementary tables of that study leave no room for question.

P3 L123 it is unclear to which results „accordance with our results“ refers to. Please revise, and or include a reference.

P9L429-431 The authors list the majority of subsites but fail to mention the “base of tongue” site that is often mentioned in the context of HPV positive HNSCC.

P11 L517- 535 the paragraph dealing with “co-morbidities” is a bit vague as co-morbidities are even more varied than HNSCC subsites. What is the cutoff for "meaningful comorbidities" and which werent recorded as trivial? Additionally, at places the authors could rephrase parts of this paragraph to mean “any co-morbidity” when referring to the “Co-morbidity Plus” subgroup of their previous study encompassing the presence of any one of the 1,152 co-morbidities recorded in those patients.

Trivial and language

Abstract line 12 and line 36 „predominantly signs responsible…“ grammatically incorrect

L96 „live-style“ typo/spelling

L132 “This notion is support by“ grammar/typo

P4 L171-173. Difficult sentence.

P6 L271 „3.4(. Fluorescence)“ punctuation problem

P7 L308 „novel blood based“ capitalization problem

P7 L313 „its pplicability“ typo

P7L 348 „HPV prevalence rate is reported to be approx. 40%.“ please specify that you refer to OPSCC HPV prevalence (if so)

P9 L413 „Thus, the by us continuously applied categorization“ difficult sentence

P9L442 „co-workers(73)“ and elsewhere spacing is inconsistent between word and reference bracket.

P9L443  since decimal point is used, the thousands separator should be a comma instead of a point to avoid confusion

P11 L514 and 518 Possibly there should be a space when listing multiple references in a single bracket. this is sometimes done but more often not

P13 L622 typo “mechanisms my eventually“

Author Response

Thank you very much for the careful reading of the manuscript!

Point by point answer to the comments (with all changes in track change modus):

P2 L50 was changed to “an important” prognostic factor.

P3 L 103-107 the passage in ( ) was removed

P3 L123 In our opinion, the sentence with the accordance is directly related to the sentence about our study (29) before it. However, we have additionally indicated the reference 29 at the desired position.

P9 L429-431 both, lingual (base of tongue) and palatine tonsils per definition are part of the Waldeyers´s tonsillar ring in terms of biological function which again is part of the anatomic region “oropharynx”. Therefore, the terminus oropharynx already includes base of tongue. In fact, however, the base of tongue histopathologically speaking is a tonsil (lymphoepithelial tissue). To clarify this, in line 413 we inserted "(base of tongue (lingual) and palatine tonsil)" following Waldeyers' tonsillar ring.

P11 L 517-535 The authors understand what the reviewer is getting at. However, the issue highlighted is challenging, as there are no cut offs or definitions of "meaningful" co-morbidity in the strict sense. However, we used various co-morbidity indeces in the study. The use of such indices as the CCI hopefully gives the reader an inkling that certain standards have been met in the evaluation of the study results. Accordingly, we have added the application of the indices in line 424 (applying various co-morbidity indices as the CCI). We hope the latter meets the concerns of the reviewer.

The majority of “Trivial and language” we have changed according to the kind comments:

We, however, need support since we don´t see the incorrect grammar in the abstract. How would it be more correct?
